# Development and validation of LC-MS/MS method for imatinib and norimatinib monitoring by finger-prick DBS in gastrointestinal stromal tumor patients

Valentina Iacuzzi[1,2☉], Bianca Posocco[1☉]*, Martina Zanchetta[1,3], Marcella Montico[4], Elena Marangon[1], Ariana Soledad Poetto[1,5], Mauro Buzzo[1], Sara Gagno[1], Angela Buonadonna[6], Michela Guardascione[1], Bruno Casetta[1,7], Giuseppe Toffoli[1]

**1** Experimental and Clinical Pharmacology Unit, Centro di Riferimento Oncologico di Aviano (CRO) IRCCS, Aviano, Italy, **2** Doctoral School in Nanotechnology, University of Trieste, Trieste, Italy, **3** Department of Chemical and Pharmaceutical Sciences, University of Trieste, Trieste, Italy, **4** Scientific Directorate, Centro di Riferimento Oncologico di Aviano (CRO) IRCCS, Aviano, Italy, **5** Doctoral School in Pharmacological Sciences, University of Padua, Padua, Italy, **6** Medical Oncology Department, Centro di Riferimento Oncologico (CRO) IRCCS, Aviano, Italy, **7** Polo Tecnologico Pordenone, Pordenone, Italy

☉ These authors contributed equally to this work.
* bposocco@cro.it

**Data Availability Statement:** All relevant data are within the manuscript.

## Abstract

The introduction of imatinib, an oral tyrosine kinase inhibitor, as first-line standard therapy in patients with unresectable, metastatic, or recurrent gastro-intestinal stromal tumor (GIST), strongly improved their treatment outcomes. However, therapeutic drug monitoring (TDM) is recommended for this drug due to the large inter-individual variability in plasma concentration when standard dose is administered. A $C_{min}$ higher than 760 ng/mL was associated with a longer progression free survival. Thus, a LC-MS/MS method has been developed and fully validated to quantify imatinib and its active metabolite, norimatinib, in finger-prick dried blood spot (DBS). The influence of hematocrit, sample homogeneity, and spot size and the correlation between finger-prick and venous DBS measurements were also assessed. The method showed a good linearity ($R^2 > 0,996$) between 50–7500 ng/mL for imatinib and 10–1500 ng/mL for norimatinib. Analytes were extracted from DBS samples by simply adding to 3 mm-discs 150 μL of acidified methanol containing IMA-D8. The collected extract was then injected on a LC Nexera system in-house configured for the on-line cleanup, coupled with an API-4000 QT. The chromatographic separation was conducted on a Synergi Fusion-RP column (4 μm, 2x50 mm) while the trapping column was a POROS R1/20 (20 μm, 2x30 mm). The total run time was 8.5 min. DBSs stored at room temperature in plastic envelopes containing a silica-gel drying bag were stable up to 16 months.

The proposed method was applied to 67 clinical samples, showing a good correlation between patients' finger-prick DBS and plasma concentrations, measured by the reference LC-MS/MS method, internally validated. Imatinib and norimatinib concentrations found in finger-prick DBS were adjusted by hematocrit or by an experimental correction factor to estimate the corresponding plasma measurements. At the best of our knowledge, the proposed

**Funding:** The authors received no specific funding for this work.

**Competing interests:** The authors have declared that no competing interests exist.

LC-MS/MS method is the first analytical assay to measure imatinib and norimatinib in DBS samples.

## Introduction

Imatinib (Gleevec® or Glivec®), hereinafter referred to as IMA, belongs to the family of tyrosine-kinase inhibitors (TKIs) (Fig 1). The introduction of this drug on the market turned out to be a revolution in cancer therapy. It is used as first line-treatment in Philadephia-positive chronic myeloid leukemia (Ph+CML) [1] and it has been approved for the use in metastatic or unresectable gastrointestinal stromal tumors (GISTs) [2] based on the increased progression-free (PFS) and overall survival (OS) [3].

For this drug, like for many other antineoplastic medicines, there is a large inter-individual variability in plasma concentration after administration of the standard dose [4]. This can induce either toxic side effects, in the case of higher plasma concentrations, or the ineffectiveness of the therapy when they are below the cutoff. Because of inter-individual variability due to differences in pharmacokinetics of IMA (absorption, distribution, metabolism and excretion), finding the optimal dose for each patient seems to be a challenge. For this reason, therapeutic drug monitoring (TDM) is recommended to maintain plasma concentrations within the targeted therapeutic window in order to maximize the efficacy and minimize the toxicity of the therapy [5].

It has been proved that there is a good relationship between IMA plasma concentration and pharmacodynamic effect (i.e. response rate, progression free/ overall survival, toxic side effects) [6]. For a complete cytogenetic or major molecular response in patients with Ph+ CML, IMA $C_{min}$ should be above 1000 ng/mL [7], and above 1100 ng/mL in GIST patients for extending the progression-free survival [4]. As related to GIST patients, Eecheoute et al. showed that the $C_{min}$ of IMA decreased by about 30% in the first three months of treatment [8]. In this study, a longer PFS was associated with $C_{min}$ values higher than 760 ng/mL, suggesting the need to repeatedly evaluate IMA plasma concentrations during the treatment period. To assess the treatment consistency, the ideal scenario should be to perform the $C_{min}$ measurement every day and for a significantly extended period [9]. These high frequency

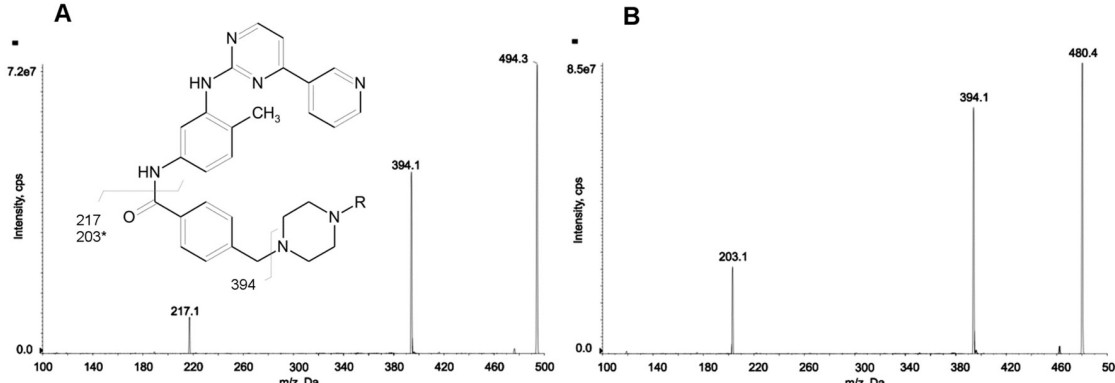

**Fig 1. MS/MS mass spectra of IMA (A) and norIMA (B) with chemical structures and identification of the main fragment ions.** MS/MS spectra were obtained with a CE = 30 V; R = CH₃ for IMA and R = H for norIMA; * referred to norIMA.

measurements should impose that specimen collection is home-made, far from hospital premises, and in a patient-friendly way.

Therapeutic drug monitoring supported by liquid chromatography coupled with tandem mass spectrometry (LC-MS/MS) is the currently used strategy in order to quantify the drug in patients. Many validated methods for performing TDM by mass spectrometry in serum or plasma have been developed [10]. Nevertheless, sample collection remains the main limitation for a routine use of TDM in clinical practice. It requires medical or nurse support, blood centrifugation and accurate sample storage before measurement. These limitations hamper most of the published TDM methodologies to be translated in a cost-effective and patient-friendly routine diagnostic tool.

Dried blood spot collection through finger prick is an attractive option in this scenario [11]. Relevant advantages are the minimally invasive procedure of a finger prick (amenable at home), the small volume to collect (suitable for children), and the stability of the analyte on filter paper (good for specimen transportation) [12].

Many assays in DBS have been reported in the literature over the previous years, only two of them including IMA [13, 14]. Both papers report the possibility to correlate the IMA concentrations found in DBS with those related to plasma, applying specific adjustments (hematocrit or correction factor). Anyway, neither the method proposed by Kralj et al. nor that one presented by Antunes et al. have taken in consideration the quantification of IMA main metabolite: N-desmethyl imatinib or norimatinib. In fact, it has been reported that norimatinib (norIMA) has similar pharmacological activity to IMA and it represents approximately 20–25% of the steady-state concentration of the original drug in GIST patients [15]. Therefore, its measurement should be desirable as well [6]. In addition, Antunes et al. presented a method with a calibration range (50–4000 ng/mL) not suitable to be applied in TDM for GIST patients since data from a study conducted on patients with metastatic GIST showed that IMA plasma levels ranged from 256 to 4582 ng/mL [16].

Hereby a LC-MS/MS method for the simultaneous quantification of IMA and norIMA in DBS samples is presented. This method has been cross-validated through a comparison between concentrations found in plasma and in DBS samples in GIST patients, both collected at the same time.

## Materials and methods

### Chemical and reagents

Imatinib mesylate, norimatinib (des-methyl imatinib, free base) and *d8*-imatinib mesylate (used as internal standard) were supplied by TRC (Toronto, Canada). Ammonium acetate, and formic acid (analytical grade) were bought from Sigma-Aldrich (Milano, Italy). Dimethyl sulfoxide (DMSO) was supplied by Alfa Aesar (Ward Hill, MA-USA). Acetonitrile and isopropanol, (both LC-MS grade) were obtained from Merck (Rome, Italy) while LC-MS grade methanol was purchased from Carlo-Erba (Milano, Italy). LC-MS grade water was in-house produced through, Milli-Q® IQ 7000, Millipore system (Merck Rome Italy).

Control human blood, used to prepare daily standard calibration curves and quality control (QC) samples, was collected in K-EDTA-Monovette of Sarstedt (Nuembrecht, Germany) and provided by the transfusion unit of the National Cancer Institute (Aviano, Italy) from healthy volunteers. Corresponding Hct values were measured at central lab with an Advia system (Siemens Medical Solutions Diagnostics Zurich, Switzerland). ET31-CHR filter papers were supplied by GE-Whatman (Little Chalfont, UK).

## Preparation of working solutions

IMA and norIMA stock solutions were prepared in DMSO at concentrations of 3 mg/mL and 0.6 mg/mL, respectively. The two stock solutions were mixed together with a 1:1 ratio (final concentrations of 1.5 mg/mL and 0.3 mg/mL for IMA and norIMA, respectively) and then further diluted in methanol 50% for getting the working solutions to be used to build the calibration curve (from A to J) and QC samples (L-low, M-medium, and H-high). The final concentrations obtained were: 750, 625, 500, 375, 250, 100, 50, 20, 10, 5 μg/mL (from A to J) and 600, 125, 25 μg/mL (QCH, M, and L) for IMA, and 150, 125, 100, 75, 50, 20, 10, 4, 2, 1 μg/mL (from A to J) and 120, 25, 5 μg/mL (QCH, M, and L) for norIMA. IMA-D8 stock solution was prepared in DMSO at concentration of 0.5 mg/mL. Intermediate solution at 10 μg/mL was prepared by diluting the stock solution with methanol. DBS and plasma extraction solutions were prepared diluting the intermediate IMA-D8 solution with methanol added with 0.1% of formic acid to obtain the final concentration of 10 ng/mL.

## Preparation of standards and quality control DBS samples

Calibrators and QC DBS samples were made by mixing 1 to 100 the working solutions with whole blood from donors. The final concentrations were: 50, 100, 200, 500, 1000, 2500, 3750, 5000, 6250, 7500 ng/mL (from J to A) and 250, 1250, 6000 ng/mL (QCL, M, H) for IMA and 10, 20, 40, 100, 200, 500, 750, 1000, 1250, 1500 ng/mL (from J to A) and 50, 250, 1200 ng/mL (QCL, M, H) for norIMA.

Calibration standards and QC samples in blood were prepared by adding 3 μL of standard or QC working solutions to 297 μL of whole blood and the solution were equilibrated for 30 min (incubation time was tested up to 4h with no differences between 30 min to 4h) at 37°C. Then, 20 μL aliquots of spiked blood were spotted on filter paper and allowed to air dry for 3 h at room temperature. With a pneumatically-activated device, supplied by Analytical S&S (Flanders NJ, USA), calibration standards and QC DBS were punched for getting a 3 mm-disc which was added by 150 μL of a IMA-D8 methanolic solution (10 ng/mL) containing 0.1% formic acid. After a 30 min gentle mixing (extraction time was tested from 30 min to 4 h with no differences between them) and a 10 min centrifugation at 16000 g and 4°C, 100 μL of supernatant were transferred to a polypropylene autosampler vial for the analysis.

## LC-MS/MS equipment and conditions

All the LC-MS/MS measurements were made by an API-4000 QT (Sciex Brugherio, Italy) coupled to a LC Nexera system (Shimadzu Milano, Italy) in-house configured for the on-line cleanup. It included an autosampler, a binary pump, a column oven and an additional pump module for the trapping step. The chromatographic separation of the analytes (5 μL injection volume) was conducted on a Synergi Fusion-RP column (4 μm, 2x50 mm) from Phenomenex (Bologna, Italy) kept in the oven at 55 °C, while the trapping column was a POROS R1/20 (20 μm, 2x30 mm) from Applied-Biosystems TF (Monza, Italy). The trapping step was conducted as follow: 1.5 min at 2 mL/min of an aqueous solution of methanol 10% containing formic acid 0.1% and 2 mM ammonium acetate. The subsequent elution step, conducted in forward flow, was obtained using the following gradient: at 0.45 mL/min starting with 10% of eluent B (mixture acetonitrile-isopropanol 80:20 containing formic acid 0.1%) and 90% of eluent A (aqueous solution of formic acid 0.1% containing 2 mM ammonium acetate). After 1.5 minute, eluent B moves up to 60% in 4 min. A cleaning step at 98% lasting 1 min is performed before re-equilibration. The total run time was 8.5 min. The retention times of IMA and norIMA were 5.50 and 5.39 min, respectively. The mass spectrometer worked in positive multireaction monitoring (MRM) mode and was equipped with a valve switching system and a

Turbo Ion Spray source operating at 500˚C. The ionspray voltage was set at 2200 V with curtain gas pressure at 25 psi, and both nebulizer gas and turbo gas pressure at 40 psi. The fragmentation patterns of each compound (Fig 1) were as follow: 494.4 > 394.3 $m/z$ (quantifier, DP 110 volts, CE 40 V) and 494.4 > 217.2 $m/z$ (qualifier, DP 110 volts, CE 35 V) for IMA; 480.4 > 394.3 $m/z$ (quantifier, DP 110 volts, CE 35 V) and 480.4 > 203.2 $m/z$ (qualifier, DP 110 volts, CE 35 V) for norIMA; 502.4 > 394.2 $m/z$ (DP 110 volts; CE 40 V) for IMA-D8 used as IS. Data processing and quantifications were performed with Analyst 1.6.3.

## Validation study

The validation study of the proposed method was conducted as required by the EMA and the FDA guidance on bio-analytical method validation [17, 18] and according to EBF recommendation on the validation of bioanalytical methods for dried blood spots [19].

**Recovery and matrix effect.** The extraction recovery was determined in quintuplicate at the three QC concentrations (L, M, and H). The peak areas of IMA and norIMA, extracted from DBS QC samples, were compared to those obtained from post-extraction spiked DBS.

The post-extraction spiked DBS, in quintuplicate at the three QC concentrations, were then compared to the same QC samples prepared in neat solution (methanol), for the matrix effect estimation. Effects of matrix endogenous components on the analytes ionization were also investigated during the development of the chromatographic method by means of the post-column infusion: a constant flow of IMA and norIMA solutions prepared in methanol (250 ng/mL), were infused by a syringe pump during the chromatographic run of an extracted blank DBS sample. The extracted DBS sample eluted from the LC column and the flow from the infusion pump were combined through a zero-dead-volume tee union and inserted into the mass spectrometer source. A variation in the signal response of the infused analyte indicates ionization enhancement or suppression.

**Limit of quantification, selectivity and linearity.** The LLOQ, concentration of the lowest standard (J), is defined as the lowest concentration that could be measured with a precision within 20%, accuracy between 80% and 120% and a signal-to-noise ratio (S/N) ≥5. The LLOQ of the present method was assessed by analysing six DBS samples at J concentration (50 and 10 ng/mL for IMA and norIMA, respectively), prepared as reported in "Preparation of standards and quality control DBS samples" section. Selectivity was proved analysing blank DBS samples using blood from six individuals, prepared according to the proposed extraction procedure and individually evaluated for interferences.

To validate the linearity, calibration curves were freshly prepared over five different working days. The LC-MS/MS peak-area ratios of each analyte/IS compared to the nominal concentrations of each standard point were plotted using a least-squares linear regression applying a weighted factor of $1/x^2$. The linearity of the standard curves was checked by calculating the Pearson's determination coefficient $R^2$ and by comparison of the true and back-calculated concentrations of the calibration standards. A minimum of eight out of ten calibration points had to meet these criteria, including the LLOQ and the highest calibrator, ULOQ: the accuracy of back-calculated concentration values of each point had to be within 85–115% of the theoretical concentration (80–120% at the LLOQ).

**Intra- and inter-day precision and accuracy.** The precision and accuracy of the presented method were evaluated by analysing six replicates of each QC sample (L, M, and H) within a single-run analysis for intra-day assessment and three replicates of each QC sample over five different working days for inter-day assessment, using standard calibration curves freshly prepared. The method precision, at each concentration, was reported as the coefficient of variation (CV%), expressing the standard deviation as a percentage of the mean calculated

concentration. The accuracy of the method was determined by expressing the mean calculated concentration as a percentage of the nominal concentration. The measured concentration for at least 2/3 of the QC samples had to be within 15% of the nominal value, in each run, and only one QC sample, at each concentration level, could be excluded.

**Stability.** The stability of IMA and norIMA was assessed by analysing QC DBS samples at the three concentrations L, M, and H during sample storage and handling procedures. The stability of the QC samples, processed as previously reported ("Preparation of standards and quality control DBS samples" section), was assessed in the autosampler by repeatedly analysing the extracts 24 and 48 h after the first injection. Long-term stability of DBS samples was assessed at the storage condition applied (in plastic envelopes containing a silica-gel drying bag at room temperature) at time intervals of 1, 2, 4 weeks and then months after preparation. Long-term stability of working solutions (methanol matrix) was assessed stored at approximately −80˚C. The two analytes were considered stable when the testing samples did not exceed 15% from the nominal concentrations at each QC concentration.

**Effect of the hematocrit, sample homogeneity and spot size.** To assess the Hct effect, three aliquots of whole blood were prepared at different Hct% values: EDTA whole blood was centrifuged (2450 g, at 4˚C for 10 min) and then proper volumes of red blood cells and plasma were combined. To cover the entire Hct% range of GIST patients (spanning from 32 to 45%), the obtained Hct% values were 29, 38.4, and 59%. QCL, QCM, and QCH samples were prepared, in triplicate, for each of the three blood aliquots with different Hct values and quantified using a DBS calibration curve prepared at fixed Hct value (38.4%). For the DBS calibration curve and QC samples preparation see the above "Preparation of standards and quality control DBS samples" section. The influence of Hct on IMA and norIMA concentrations was then determined as the accuracy and precision % of the measured concentrations in DBS samples.

Due to the "volcano effect" the analyte concentration can be lower in the center than in the peripheral area of the spot. Thus, to assess whether there is a difference in IMA or norIMA concentrations between central and peripheral areas of the DBS, QCL, QCM, and QCH samples were prepared in triplicate, as reported in the above "Preparation of standards and quality control DBS samples" section. Then, the punch was performed in the center and in the edge area of the DBS for each QC sample. The homogeneity was calculated as the center/perimeter concentration ratio (C/P) and the expecting result is C/P = 1.

The influence of spot size was also investigated. QCL, M, and H samples were prepared and the spots were performed with four different blood volumes: 10, 20, 30, 40 μL, in triplicate for each QC concentration (L, M, and H). The quantification of these QC samples was performed using a DBS calibration curve made by fixed 20 μL-volume spots. The influence of spot size on IMA and norIMA concentrations was then determined as the accuracy and precision % of the measured concentrations in DBS samples.

**Plasma samples analysis.** Calibration standards and QC samples in plasma were prepared as follow: 3 μL of standard or QC working solutions were added to 297 μL of pooled plasma and vortexed for 10 s. Five microliters of spiked or patients' plasma sample were precipitated with 245 μL of the extraction solution (10 ng/mL of IMA-D8). The mixture was vortexed for 10 s and then centrifuged for 10 min at 16000 g and 4˚C. After that, 200 μL of the supernatant were transferred to a polypropylene autosampler vial for the subsequent analysis. The quantification was conducted with a LC-MS/MS method in-house developed and validated. The chromatographic separation was obtained with a Synergi Fusion-RP column (4 μm, 80 Å, 50 x 2.0 mm) coupled with a Fusion-RP security-guard pre-column (4 x 2.0 mm) (Phenomenex) under gradient condition using 0.1% formic acid/bidistilled water (v/v) and methanol/isopropanol (9:1, v/v) with 0.1% formic acid (v/v). The quantification was conducted using the following quantifier transitions: IMA $m/z$ 494.4 > 394.2, norIMA $m/z$ 480.4 > 394.2 and IS $m/z$

502.4 > 394.2. The method was linear over the concentrations range of 30–7500 ng/mL for IMA and 6–1500 ng/mL for norIMA.

**Application of the method to clinical samples.** The proposed method was applied to a clinical study for the quantification of IMA and norIMA $C_{min}$ in GIST patients (EudraCT N: 2017-002437-36, protocol code: CRO-2017-19). Patients with histologically or cytologically confirmed diagnosis of GIST and eligible for IMA treatment, age ≥18, Eastern Cooperative Oncology Group (ECOG) performance status of 0 or 1, life expectancy > 3 months were enrolled according to the routine clinical practice criteria either in adjuvant or first-line setting. Exclusion criteria were: pregnancy status; refusal of informed consent; non-collaborative and/or unreliable patients; inability to attend periodic clinical check-ups.

Patients were asked to periodically (every two months) collect DBS and plasma samples for the estimation of IMA and norIMA $C_{min}$ during therapy treatment. They were treated with different doses (range 200–600 mg/die) of IMA and they should have taken IMA pill daily at a fixed time (usually around 12.00 pm). Since timing is crucial for an accurate estimation of the real $C_{min}$, venous plasma and finger-prick have been collected immediately before the scheduled pill intake. Venous blood was drawn through 4.9 mL K-EDTA-Monovette of Sarstedt (Nuembrecht, Germany) and plasma was obtained by centrifugation for 10 minutes at 2450 g at 4˚C and transferred to polypropylene micro-tubes and stored at -20˚C until the day of analysis. Additionally, 1 mL of fresh blood was taken with a syringe (without anticoagulant) at the moment of venous sampling and drops of 20 µL volume were lied down on a piece of Whatman ET31CHR filter paper to be used as DBS controls. DBS collection through finger-prick was conducted after adequate training of the personnel. The prick was performed by a sterile lancet Securlancets from Menarini Diagnostics (Firenze, Italy) after washing the hand with warm water. Following disinfection, the first droplet was discarded while the following ones (usually two droplets) were let to fall on the filter paper.

Both the DBSs (from venous blood and from finger-prick) were let to dry at ambient air for at least 3 h and cards were stored until measurement at room temperature in plastic envelopes containing a silica-gel drying bag. The processing methods for DBS and plasma samples are reported in "Preparation of standards and quality control DBS samples" and "Plasma samples analysis" sections, respectively.

**Ethics statement regarding human samples.** The study (EudraCT number: 2017-002437-36) was approved by the local ethics committee (Comitato Etico Unico Regionale- C. E.U.R.) and by Agenzia Italiana del Farmaco (AIFA, Rome, Italy). It was conducted according to the principles expressed in the Declaration of Helsinki. Patients were informed by the oncologist about the clinical study during their visits and were recruited only after the signature of the informed consent.

**Correlation between finger-prick and venous collection.** As reported by some authors, there has been a concern whether the drug concentration after a finger-prick sampling was comparable with venous collection [20, 21]. The clinical protocol was indeed designed to collect DBS from both finger-prick and venous blood (without anticoagulant) from the same patient to compare the measurements obtained from each matrix. The comparison was performed in a subset of 10 patients' samples, estimating both the Pearson's coefficient of determination ($R^2$) and the percentage difference between the concentrations found in finger-prick ($C_{FP}$) and venous ($C_v$) DBS (calculated as: ($C_{FP}$-$C_v$)*100/mean).

**Cross-validation study.** A set of 55 DBS samples and their corresponding plasma samples were analysed to obtain the concentration of both IMA and norIMA in each matrix. A minimum sample size was initially calculated based on Lin's concordance correlation coefficient ($\rho c$). We hypothesize that the $\rho c$ between the measurement obtained from DBS sample (comparator) and that obtained from plasma samples would be 0.97 (H1). Considering a null

hypothesis (H0) of a ρc of 0.90 and an alpha level of 0.05, a sample of 19 units resulted in 0.82 power to determine a difference between H0 and H1 for both IMA and norIMA. This sample size was further increased up to 55 units in order to apply Passing-Bablok regression and Bland-Altman method, as suggested by Passing and Bablok [22], to obtain reliable results.

The estimated plasma concentration ($EC_{pla}$) was obtained from the DBS measurement ($C_{DBS}$) applying the following equation: $EC_{pla} = C_{DBS}/[1-(Hct/100)]$. Furthermore, the $EC_{pla}$ was calculated also from $C_{DBS}$ multiplied for a correction factor ($F_c$), as previously proposed by Antunes et al. [14]. The correction factor was calculated as the mean ratio between the concentrations measured in plasma and $C_{DBS}$ in all the samples analysed. This strategy allows to avoid the use of Hct correction. Agreement between the two methods (plasma and DBS) was evaluated through: a) Passing-Bablok regression: intercept and slope of the regression equation are reported with relative 95% Confidence Interval (95% CI); b) Bland-Altman method: spearman correlation coefficient (r) between differences in means and mean of the measurements is reported, for the sake of simplicity the tested measure is named Y in the text, while the plasma value (reference) is indicated as X, c) Lin's concordance correlation coefficient (LCCC), and d) FDA/EMA requirements (67% of samples tested within ±20% of the mean). Statistical analyses were performed with Stata 14.2 (StataCorp, Texas USA). As further validation, the $F_c$ calculated was then applied to a set of 12 extra samples for both IMA and norIMA.

**Incurred sample reanalysis.** Incurred Sample Reanalysis (ISR) is a very important component of the bioanalytical method validation as additional measure of assay reproducibility. ISR is conducted by repeating the analysis of a subset of patients' samples in separate runs on different days. Therefore, a set of 10 patients' DBS samples were re-analysed in a further analytical session. According to FDA and EMA guidelines, the two analyses can be considered equivalent if the percentage difference [expressed as: (repeat-original)*100/mean] between the first concentration and the second concentration measured is within ±20% for at least 67% of the samples.

## Results and discussion

### Recovery and matrix effect

According to the proposed method, analytes were extracted from DBS samples by simply adding to 3 mm-disc 150 μL of methanol added with 0.1% of formic acid and IMA-D8 as internal standard at the concentration of 10 ng/mL. The recovery resulted in the range 74.8–80.5% with a CV ≤5.4% for IMA and 66.5–68.5% with a CV ≤7.7% for norIMA, as shown in Table 1. In the same Table, the estimated matrix effect (ME%) is reported: it was found between 90.0–109.7% with a CV within 6.4% for IMA and between 100.5–110.1% with a CV within 7.2% for norIMA, indicating the absence of major ion suppression or enhancement for both the analytes. These results were also confirmed by the post column infusion test: no significant

**Table 1. Recovery and matrix effect (ME) of IMA and norIMA in DBS samples.**

| Analyte | Nominal conc. (ng/mL) | Recovery (%) ±SD | Recovery CV (%) | ME (%) ±SD | ME CV (%) |
|---|---|---|---|---|---|
| IMA | 250 | 74.8±4.1 | 5.4 | 109.7±4.4 | 4.0 |
| | 1250 | 77.7±303 | 4.2 | 104.8±4.9 | 4.7 |
| | 6000 | 80.5±2.0 | 2.5 | 90.0±5.7 | 6.4 |
| norIMA | 50 | 66.5±5.1 | 7.7 | 109.6±1.9 | 1.8 |
| | 250 | 66.6±3.1 | 4.6 | 110.1±1.6 | 1.5 |
| | 1200 | 68.5±1.4 | 2.1 | 100.5±7.2 | 7.2 |

variations in the signal of IMA and norIMA were detected at the retention times of the two analytes.

## Limit of quantification, selectivity and linearity

The LLOQ for the proposed method was fixed at 50 ng/mL for IMA and 10 ng/mL for nor-IMA. As shown in Fig 2, the S/N ratio was 149 for IMA and 16 for nor-IMA. The accuracy and CV% were, respectively, 97.1 and 7.6% for IMA and 91.1% and 8.7% for norIMA.

The method has a good selectivity: from the analysis of six blank DBS samples no significant interferences were detected, especially at the retention times of our compounds. In Fig 2, an example of one of the six blank DBS samples analysed is reported.

Good linearity was obtained over the concentration range of 50–7500 ng/mL for IMA and 10–1500 ng/mL for norIMA, being the Pearson's coefficient of determination $R^2 \geq 0.9963$ for each run. In Table 2, complete data ($R^2$, intercept, and slope) related to linearity of both IMA and norIMA calibration curves are reported. The accuracy resulted in the range 91.4–108.0% for IMA and 91.4–105.7% for norIMA and the precision, expressed as CV%, was within 8.9% for IMA and within 8.4% for norIMA.

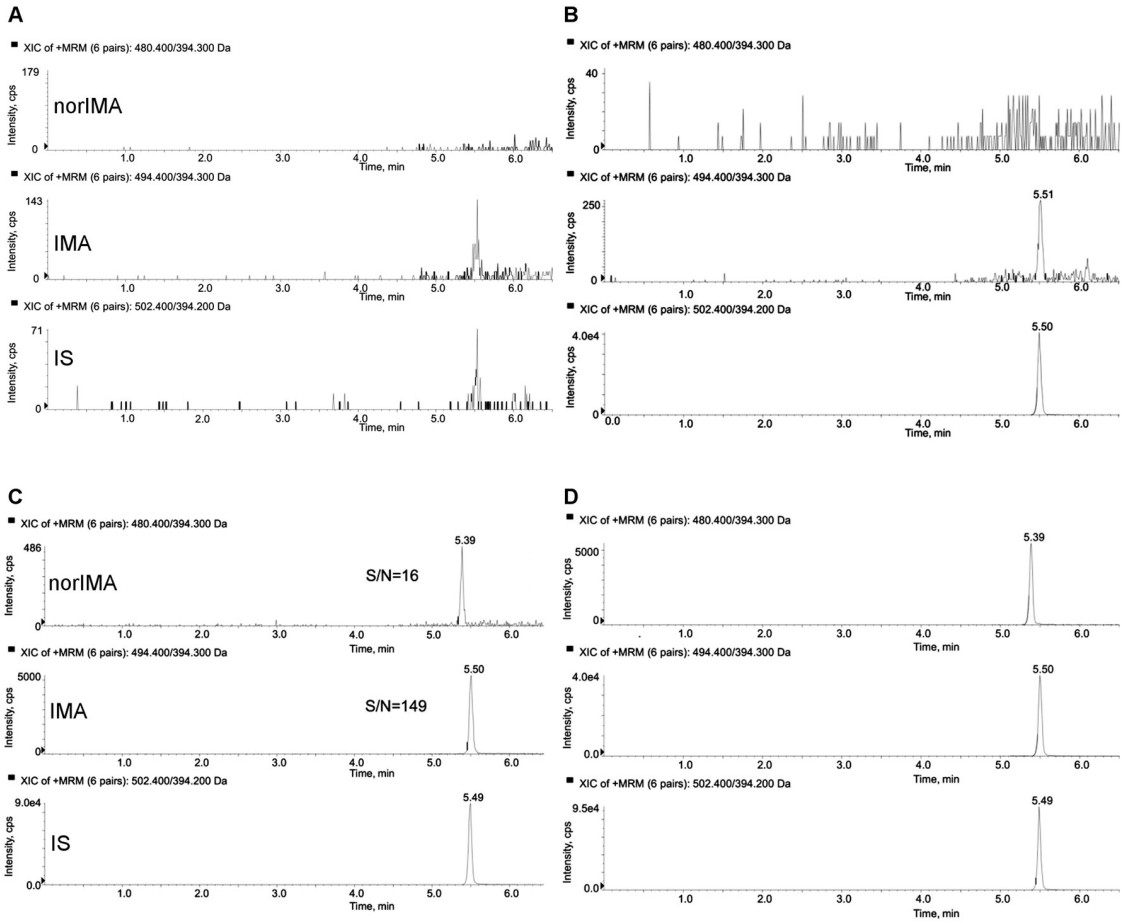

**Fig 2. Representative SRM chromatograms.** (A): blank DBS sample; (B): blank DBS sample with IS added; (C): S/N of IMA and norIMA at the LLOQ (50 and 10 ng/mL, respectively); (D): extracted DBS sample of a treated patient showing IS, IMA (575 ng/mL) and norIMA (94.3 ng/mL).

**Table 2. Linearity, accuracy and precision data for calibration curves of IMA and norIMA.**

IMA
$R^2$ = 0.9974±0.0007
Intercept: 0.0201±0.004
Slope: 0.00087±0.00006

| nominal conc. (ng/mL) | Mean ± SD | CV% | Acc% |
|---|---|---|---|
| 50 | 49.2±2.3 | 4.7 | 98.5 |
| 100 | 101.0±9.0 | 8.9 | 101.0 |
| 200 | 211.2±10.6 | 5.0 | 105.6 |
| 500 | 538.2±17.6 | 3.3 | 107.6 |
| 1000 | 1080.4±41.2 | 3.8 | 108.0 |
| 2500 | 2457.4±115.4 | 4.7 | 98.3 |
| 3750 | 3848.4±185.9 | 4.8 | 102.6 |
| 5000 | 4825.8±195.4 | 4.0 | 96.5 |
| 6250 | 5772.0±318.2 | 5.5 | 92.4 |
| 7500 | 6856.0±303.3 | 4.4 | 91.4 |

norIMA
$R^2$ = 0.9970± 0.0009
Intercept: 0.0017± 0.0007
Slope: 0.00037±0.00005

| nominal conc. (ng/mL) | Mean ± SD | CV% | Acc% |
|---|---|---|---|
| 10 | 9.9±0.8 | 8.4 | 98.7 |
| 20 | 21.1±1.6 | 7.6 | 105.3 |
| 40 | 41.5±2.2 | 5.2 | 103.8 |
| 100 | 105.7±5.3 | 5.0 | 105.7 |
| 200 | 211.2±7.9 | 3.7 | 105.6 |
| 500 | 489.5±30.7 | 6.3 | 97.9 |
| 750 | 780.2±30.4 | 3.9 | 104.0 |
| 1000 | 980.7±54.9 | 5.6 | 98.1 |
| 1250 | 1170.7±76.2 | 6.5 | 93.7 |
| 1500 | 1370.3±76.2 | 5.6 | 91.4 |

## Intra- and inter-day precision and accuracy

The precision of the method was confirmed by the intra- and inter-day CV ≤3.1% and ≤5.6% for IMA and ≤4.3% and ≤6.6% for norIMA (Table 3). The intra- and inter-day accuracy were within the range 88.9–106.2% and 98.9–104.3% for IMA and 92.9–112.8% and 95.7–101.0% for norIMA.

## Stability

Both IMA and norIMA resulted stable, after extraction from DBS, for 24 h at 4˚C. The long-term stability in DBS, stored in plastic envelopes containing silica-gel drying bag at room temperature, was verified up to 16 months. The stability of the standard working solutions of IMA and norIMA was assessed after 24 months of storage at −80˚C.

## Effect of the hematocrit, sample homogeneity and spot size

No significant impact of Hct on IMA and norIMA quantification in DBS was observed among the Hct values range tested (29.0–59.0%). Accuracy was between 95.3 and 104.8% for IMA and between 95.4 and 108.9% for norIMA at each QC level (L, M, and H). Precision was within 3.5 and 6.0% for IMA and norIMA, respectively.

**Table 3. Intra and inter-day precision and accuracy of the method for the quantification of IMA and norIMA.**

| Intra-day (N = 6) | | | | |
|---|---|---|---|---|
| Analyte | Nominal conc. (ng/mL) | Mean ± SD | CV% | Acc% |
| IMA | 250 | 281.2 ± 6.3 | 2.2 | 88.9 |
| | 1250 | 1402.5 ± 43.5 | 3.1 | 89.1 |
| | 6000 | 5651.7±163.5 | 2.9 | 106.2 |
| norIMA | 50 | 52.1 ± 2.3 | 4.3 | 95.9 |
| | 250 | 269.2 ± 7.7 | 2.9 | 92.9 |
| | 1200 | 1064.0±26.08 | 2.5 | 112.8 |
| Inter-day (N = 15) | | | | |
| Analyte | Nominal conc. (ng/mL) | Mean ± SD | Precision % | Accuracy % |
| IMA | 250 | 260.9 ± 11.2 | 4.3 | 104.3 |
| | 1250 | 1282.3 ± 41.7 | 3.3 | 102.6 |
| | 6000 | 5932.8 ± 330.1 | 5.6 | 98.9 |
| norIMA | 50 | 50.5 ± 2.9 | 5.7 | 101.0 |
| | 250 | 248.8 ± 16.4 | 6.6 | 99.5 |
| | 1200 | 1196.3 ± 72.2 | 6.0 | 95.7 |

As related to the "volcano effect" assessment, no differences in IMA or in norIMA concentrations between central and peripheral areas of the DBS were observed. As reported in Table 4, at each concentration tested (L, M, and H) the C/P was within 0.9–1.1.

Finally, no influence of spot size among 10–40 μL-volume range was obtained at each QC concentration. The accuracy was between 90.0–110.1% and 90.1–104.3% for IMA and norIMA, respectively. The precision was within 6.0% and 6.5% for IMA and norIMA, respectively.

## Application of the method to clinical samples

The proposed method was applied to quantify, for the first time, both IMA and norIMA in DBSs collected from GIST patients enrolled in the clinical study from February 2018 to February 2019. 55 DBS samples were collected from 26 patients with a mean of two $C_{min}$ evaluations for each patient. Patient population was characterized as reported in Table 5. Taking into account all 55 samples, the mean Hct% was of 37.7 (range 26.2–44.1), patients were receiving IMA at the dose of: 400 mg/die in 51 samples, 200 mg/die in 1 sample, 300 mg/die in 2 samples, and 600 mg/die in 1 sample. Samplings were mostly performed 25 h after the last pill intake (25.1 ± 6.7 h).

The correlation between $C_{min}$ values from finger-prick and those from venous (without anticoagulant) DBSs, collected from ten patients, demonstrated that the two samples were equivalent ($R^2$ = 0.9967 for IMA and $R^2$ = 0.9798 for norIMA). Moreover, the percentage

**Table 4. Center/Perimeter concentration ratio (C/P) for IMA and norIMA as evaluation of the "volcano effect".**

| Sample | Nominal conc. (ng/mL) | Mean central conc. ± SD (ng/mL) | Mean peripheral conc. ± SD (ng/mL) | C/P |
|---|---|---|---|---|
| QCL (IMA) | 250 | 258,5±2,6 | 253,9±2,4 | 1.0 |
| QCM (IMA) | 1250 | 1249,7±4,6 | 1321,6±1,8 | 0.9 |
| QCH (IMA) | 6000 | 5934,4±3,1 | 5844,4±1,8 | 1.0 |
| QCL (norIMA) | 50 | 48,6±4,3 | 44,9±1,1 | 1.1 |
| QCM (norIMA) | 250 | 225,9±3,6 | 243,0±2,2 | 0.9 |
| QCH (norIMA) | 1200 | 1139,1±2,3 | 1134,0±0,8 | 1.0 |

**Table 5. Demographic and clinical characteristics of enrolled patients.**

| Patients characteristic | N |
| --- | --- |
| Sex | 14 female<br>12 male |
| Age (range) | 66 (37–83) years |
| Setting | 6 adjuvant<br>20 metastatic |
| Primary tumor site | 10 gastric<br>16 non gastric |

difference between DBSs from venous blood and those from finger-prick were always within ±20% of the mean (from -12 to 3.8%, Fig 3). This agreement may be due to the fact that the $C_{min}$ sampling, by its own definition, was performed once distribution equilibrium of the drug was reached, while differences can be expected in the early moments following drug administration [23].

## Cross-validation study

IMA concentrations found in DBS were on average 59±6% of those obtained from plasma, while norIMA concentrations from DBS samples were on average 63±9% of those from plasma. Among the 55 samples analyzed, IMA concentrations range from 310 to 5840 ng/mL in plasma and from 183 to 3340 ng/mL in DBS, while norIMA concentrations range from 67 to 672 ng/mL in plasma and from 45 to 409 ng/mL in DBS. In one out of 55 samples IMA and norIMA concentrations were found <LLOQ, in both plasma and DBS matrices. The intercept values for Passing-Bablok regression were 32.5 (95% CI: 2.9–66.0) and 1.2 (95% CI: -8.8–13.2) for IMA and norIMA, respectively, suggesting a small constant error for both analytes. The slope coefficients were 0.55 (95% CI: 0.52–0.58) and 0.61 (95% CI: 0.57–0.67) for IMA and norIMA respectively, indicating a proportional error (the higher concentrations were underestimated). This pattern was confirmed also by Bland-Altman analyses (Spearman correlation between (Y-X) and (X+Y)/2: r = -0.9449).

To estimate the plasma concentration of IMA and norIMA starting from the measurements of DBS samples we applied the equation $EC_{pla} = C_{DBS}/[1-(Hct/100)]$, according to Kralj et al. [13]. Contrary to Antunes et al. [14], we did not take into account the plasma to blood partition ($f_p$) since the value of $f_p$ we calculated according to Antunes et al. [24] was 0.45 and 0.48 for IMA and norIMA respectively, thus indicating almost complete amount of these compounds in plasma. A good agreement between the two methods (plasma and DBS normalized by Hct) was obtained according to EMA/FDA guidelines: 89% (49/55) and 78% (43/55) of $EC_{pla}$ resulted within ±20% of the mean for IMA and norIMA, respectively. Also, with the Hct normalization, the intercept values for Passing-Bablok regression (77.1, 95% CI: 1.7–181.0 and 10.5, 95% CI: -5.6–31.4, for IMA and norIMA, respectively) indicated that the small constant error is maintained for both analytes (Fig 4). Anyway, the slope coefficients were, in this case, very close to 1 (0.86, 95% CI: 0.77–0.94, and 0.97, 95% CI: 0.89–1.06 for IMA and norIMA, respectively) indicating a good agreement between the measurements and that the initial proportional error is absent or negligible. Bland-Altman analyses confirmed these results, with a Spearman correlation between (Y-X) and (X+Y)/2 of r = -0.4807 for IMA and r = -0.1132 for norIMA.

The $F_c$ calculated was 1.73 for IMA and 1.61 for norIMA. The agreement between the two methods was higher with the application of the $F_c$ than with the Hct normalization. As related to EMA/FDA guidelines requirements, 93% (51/55) and 85% (46/55) of $EC_{pla}$ resulted within

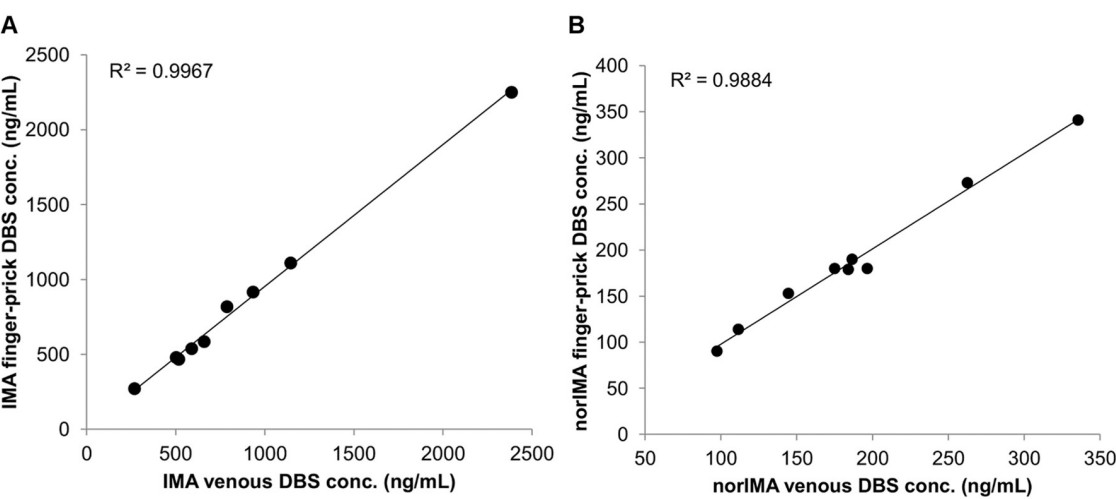

**Fig 3. Correlation between finger-prick and venous collection.** Correlation obtained for IMA (A) and norIMA (B) by comparing the results on DBS from venous blood (without anticoagulant, x-axes) and DBS from finger-prick (y-axes).

±20% of the mean for IMA and norIMA, respectively. The slope coefficients of Passing-Bablok regression were nearly equal to 1 for both analytes (0.96, 95% CI: 0.91–1.01, and 1.00, 95% CI: 0.92–1.09 for IMA and norIMA, respectively), while no constant errors resulted for both the analytes from the intercept values (53.3, 95% CI: 0.0–107.4 and 1.8, 95% CI: 0.9–1.1 for IMA and norIMA, respectively). The LCCC (Lin's Concordance Correlation Coefficient) calculated were very high for both Hct and $F_c$ normalization methods, with a little improvement for IMA with $F_c$ correction respect to Hct one (IMA LCCC = 0.9667 and 0.9850 after Hct and $F_c$ normalization, respectively, norIMA LCCC = 0.9698 and 0.9742 after Hct and $F_c$ normalization, respectively).

A further validation of this strategy was assessed through the application of the $F_c$ of both IMA and norIMA to 12 extra patients' samples. The agreement between concentrations in plasma and in DBS samples, after the $F_c$ normalization, was verified: 100% of $EC_{pla}$ resulted within ±20% of the mean for both IMA (from -13.9 to 13.8%) and norIMA (from -7.0 to 18.7%), with a good linearity with plasma concentrations ($R^2$ = 0.9895 for IMA and $R^2$ = 0.9474 for norIMA, Fig 5).

### Incurred sample reanalysis

To further assessed the reproducibility of the proposed method, 10 DBS patients' samples were analyzed two times, with separate runs and during different working days. The concentration range of the chosen DBS samples was between 195–1700 ng/mL (quantified with the first analysis). The percentage differences obtained between the first and the second analysis were in all 10 samples within ±20%: from -16 to 1.9% for IMA and from -12.2 to 12.7% for norIMA. In Fig 6 the correlation graphs between the IMA and norIMA concentrations calculated with the first analysis against the second one are reported, showing a good linearity between the two quantifications ($R^2$ = 0.9859 for IMA and $R^2$ = 0.9824 for norIMA).

### Conclusion

At the best of our knowledge, for the first time a fast and sensitive LC-MS/MS method was developed for the simultaneous quantification of IMA and norIMA in finger-prick DBS samples. This method was successfully validated and applied to quantify finger prick DBS samples

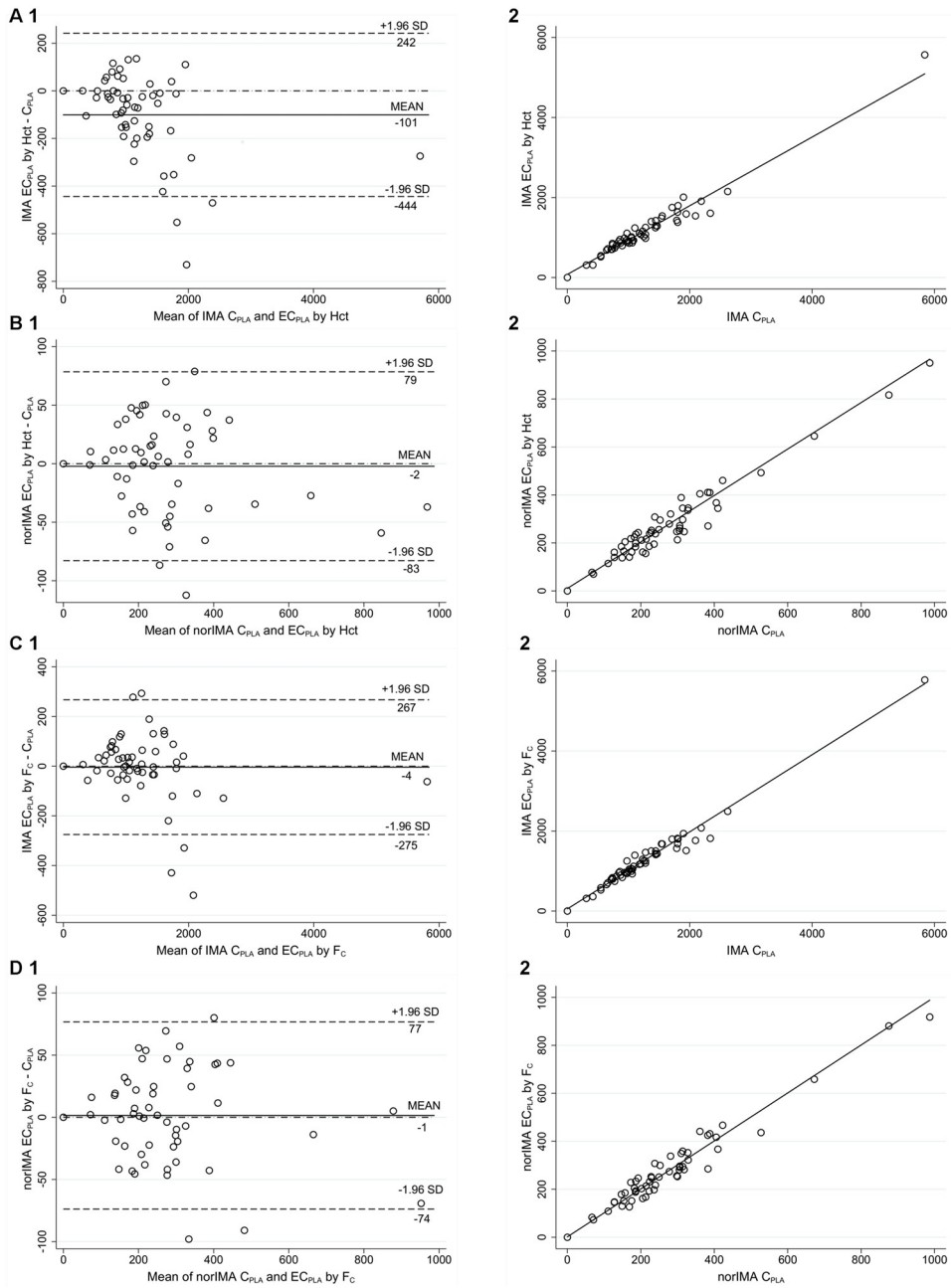

**Fig 4. Correlation between IMA and norIMA concentrations in DBS samples after Hct and $F_c$ normalization and those obtained from plasma samples.** Comparison between IMA (A) and norIMA (B) $EC_{pla}$ by Hct with corresponding $C_{PLA}$ through Bland-Altman plot (1) and Passing-Bablok regression (2). Comparison between IMA (C) and norIMA (D) $EC_{pla}$ by $F_c$ with corresponding $C_{PLA}$ through Bland-Altman plot (1) and Passing-Bablok regression (2). $EC_{pla}$: estimated plasma concentration by normalization of DBS measurements; $C_{PLA}$: concentration found in plasma samples; Hct: hematocrit; $F_c$: correction factor. All the concentrations are expressed as ng/mL. N = 55.

from patients enrolled in a clinical study. Good agreement was obtained between IMA and norIMA concentrations found in DBS and plasma samples, collected at the same time, applying the Hct normalization. Moreover, we demonstrated the possibility to avoid the Hct normalization, by simply multiplying the DBS concentration with a correction factor, not only for

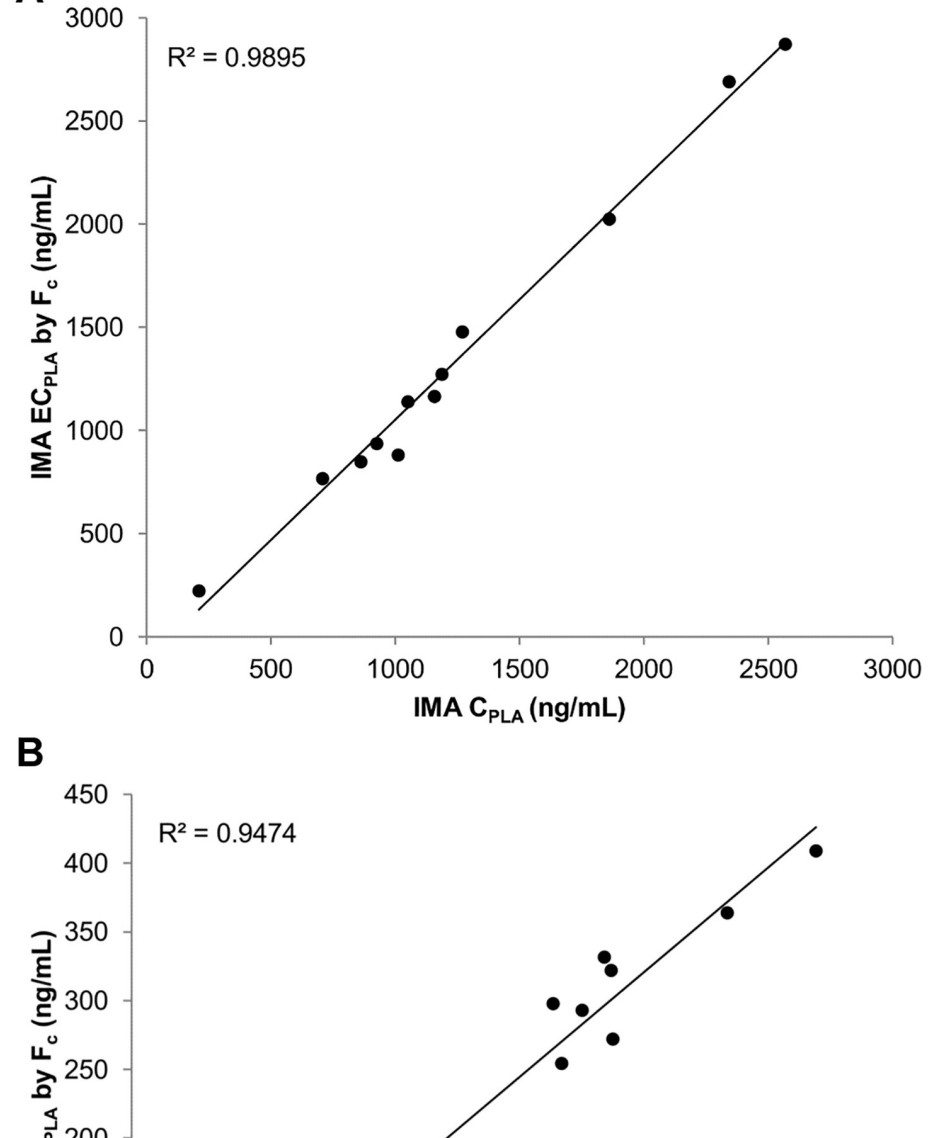

**Fig 5. Fc application to 12 external DBS samples.** Correlation graph between (A) IMA and (B) norIMA estimated plasma concentrations ($EC_{PLA}$) by Fc and their corresponding plasma measurements ($C_{PLA}$); N = 12.

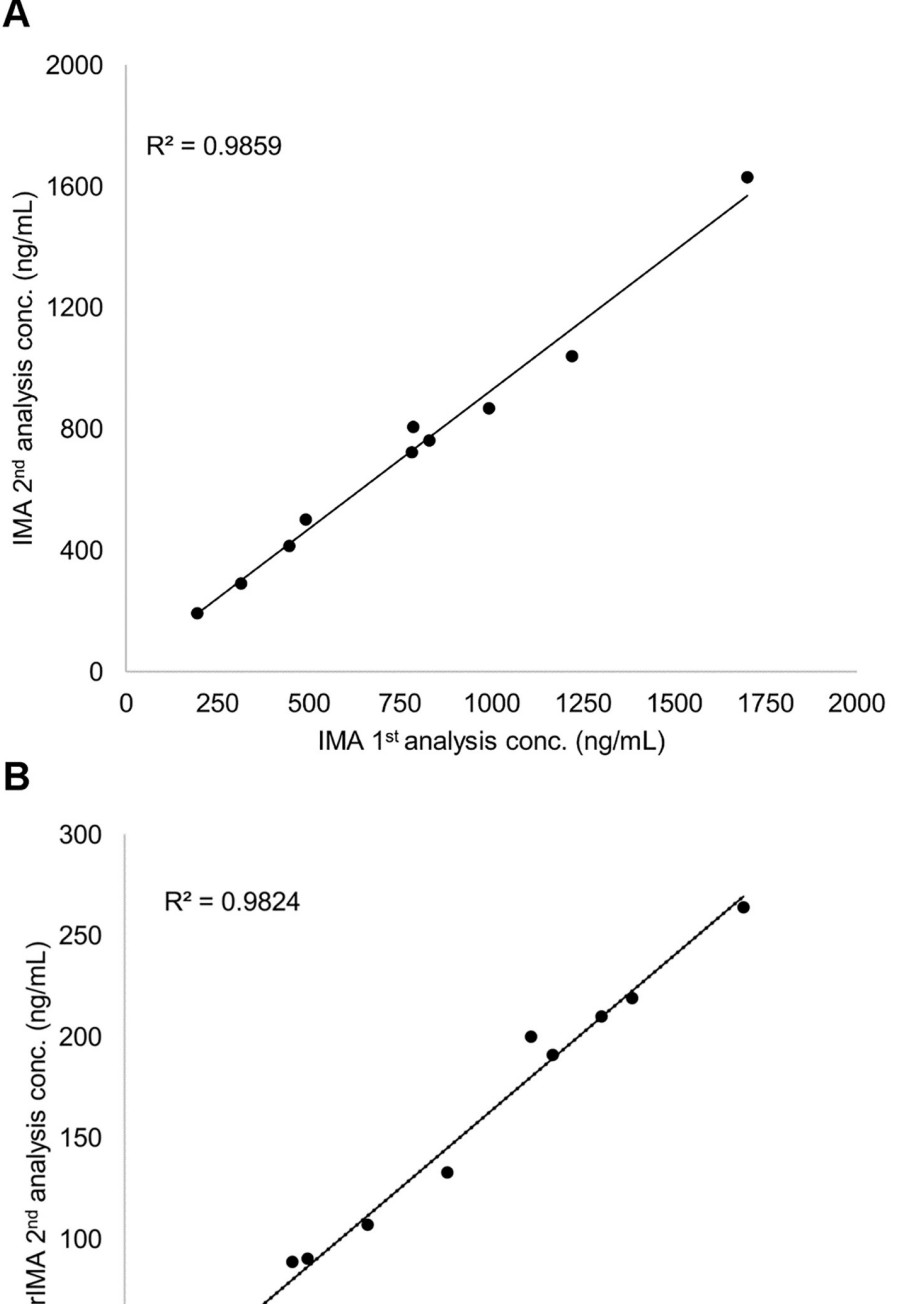

**Fig 6. Incurred sample reanalysis.** Correlation graph between the first and the second analysis of IMA and norIMA in DBS patients' samples; N = 10.

IMA, as previously reported, but also for norIMA. The correction factors applied were calculated on the basis of 55 patients samples and successively tested in a cohort of additional 12 patients' samples. The statistical analyses demonstrated that the application of the correction factor even augmented the agreement with the plasma concentrations, especially for IMA.

We hope that the possibility to perform TDM of IMA and its main metabolite in finger prick DBS samples, by simply applying a correction factor or by normalizing by Hct to obtain the plasma concentrations, will actually improve its application to clinical practice.

## Acknowledgments

We thank the patients for their participation in the clinical study.

## Author Contributions

**Conceptualization:** Bruno Casetta.

**Formal analysis:** Marcella Montico.

**Investigation:** Valentina Iacuzzi, Bianca Posocco, Martina Zanchetta, Ariana Soledad Poetto, Mauro Buzzo, Sara Gagno.

**Methodology:** Elena Marangon, Bruno Casetta.

**Resources:** Angela Buonadonna, Michela Guardascione.

**Supervision:** Giuseppe Toffoli.

**Validation:** Elena Marangon.

**Writing – original draft:** Valentina Iacuzzi, Bianca Posocco.

**Writing – review & editing:** Bianca Posocco, Giuseppe Toffoli.

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
