## [Decision Letter · Decision Letter 0]

21 Aug 2019

PONE-D-19-17238

Development and Validation of LC-MS/MS Method for Imatinib and Norimatinib Monitoring by Finger-prick DBS in Gastrointestinal Stromal Tumor Patients

PLOS ONE

Dear Dr. Posocco,

Thank you for submitting your manuscript to PLOS ONE. After careful consideration, we feel that it has merit but does not fully meet PLOS ONE’s publication criteria as it currently stands. Therefore, we invite you to submit a revised version of the manuscript that addresses the points raised during the review process.

We would appreciate receiving your revised manuscript by Oct 05 2019 11:59PM. To enhance the reproducibility of your results, we recommend that if applicable you deposit your laboratory protocols in protocols.io, where a protocol can be assigned its own identifier (DOI) such that it can be cited independently in the future. For instructions see: http://journals.plos.org/plosone/s/submission-guidelines#loc-laboratory-protocols

We look forward to receiving your revised manuscript.

Kind regards,

John Matthew Koomen, PhD

Academic Editor

PLOS ONE

Journal Requirements:

2.  In your Methods section, please provide additional information about the participant recruitment method and the demographic details of your participants. Please ensure you have provided sufficient details to replicate the analyses such as: a) the recruitment date range (month and year), b) a description of any inclusion/exclusion criteria that were applied to participant recruitment, c) a table of relevant demographic details, and d) a description of how participants were recruited.

3. Please provide a justification of the sample size used in this study.

Reviewers' comments:

Reviewer's Responses to Questions

**Comments to the Author**

1. Is the manuscript technically sound, and do the data support the conclusions?

Reviewer #1: Yes

Reviewer #2: Yes

2. Has the statistical analysis been performed appropriately and rigorously? 

Reviewer #1: Yes

Reviewer #2: Yes

3. Have the authors made all data underlying the findings in their manuscript fully available?

Reviewer #1: Yes

Reviewer #2: Yes

4. Is the manuscript presented in an intelligible fashion and written in standard English?

Reviewer #1: Yes

Reviewer #2: Yes

5. Review Comments to the Author

Reviewer #1: Minor editing for English grammar and usage. Line 104 should have a reference. Which model Millipore was used to produce water in line 135. Working solutions and calibrators are claimed to be made "from A to L", however only ten working solutions and calibrators were identified (Lines 150, 152, 161 & 163). Line 345 states that the clinical protocol was designed to collect blood "without anticoagulant", however lines 319 & 320 state that in the clinical protocol, venous blood was collected with K-EDTA, an anti-coagulant. Was a separate vacutainer collected for the correlation between finger prick and venous collection? Line 505, indicate "LCCC" stands for Lin’s Concordance Correlation Coefficient. Typo on Line 542 "LC-M/MS"

Reviewer #2: I think the article is well written in a straightforward manner, thus easy to follow by the readers. The validation experiments done by the authors are comprehensive and sufficient.

There are only a few minor revisions/addendum:

L152, 159-163: There is some error in listing of quality control concentrations, the final concentrations do not add up from the working concentrations. It is suggested to keep the order of concentrations consistent, best listed in an ascending manner.

L301-304: The HPLC parameters changed from the method development (L186-189) to the plasma sample analysis. Please discuss why it has changed and the effects it brought. Normally, the method validated should be kept the same in GLP conditions.

L449: change “follow” to “follows”

6. PLOS authors have the option to publish the peer review history of their article (what does this mean?). If published, this will include your full peer review and any attached files.

Reviewer #1: Yes: Samer Sansil

Reviewer #2: No

---

## [Author Response · Author response to Decision Letter 0]

2 Sep 2019

Reviewer #1:

Minor editing for English grammar and usage.

The manuscript has been revised for English grammar.

Line 104 should have a reference.

As properly suggested by Reviewer, a reference has been added to line 104 related to the advantages of DBS in TDM practice.

Which model Millipore was used to produce water in line 135.

The Millipore model used was Milli-Q® IQ 7000. This information has been added to the manuscript (Line 130).

Working solutions and calibrators are claimed to be made "from A to L", however only ten working solutions and calibrators were identified (Lines 150, 152, 161 & 163).

The Reviewer is perfectly right since the Italian alphabet was mistakenly used to name the working solutions (thus from A to L we have only 10 letters). The manuscript has been revised according to the English alphabet (thus the working solutions are now named from A to J).

Line 345 states that the clinical protocol was designed to collect blood "without anticoagulant", however lines 319 & 320 state that in the clinical protocol, venous blood was collected with K-EDTA, an anti-coagulant. Was a separate vacutainer collected for the correlation between finger prick and venous collection?

The Reviewer is right since the description of samples collection (Line 318-328) was erroneously reported. In fact, 1 mL of venous blood was collected without anticoagulant using a syringe and directly transferred into 1.5 mL tube. The blood was immediately spotted with a Gilson pipette on a piece of Whatman ET31CHR filter paper to make Control – DBS (the spots were 20 µL in volume). This step needed to be fast to avoid blood coagulation inside the tubes. The chapter has been properly corrected.

Line 505, indicate "LCCC" stands for Lin’s Concordance Correlation Coefficient.

The LCCC abbreviation was introduced at the first mention of Lin’s concordance correlation coefficient (Line 373). Anyway it could be easier for the reader to make explicit what LCCC stands for also in Line 505 since it is an uncommon abbreviation. 

Typo on Line 542 "LC-M/MS"

We thank the reviewer for finding this typo that has been corrected.

Reviewer #2:

I think the article is well written in a straightforward manner, thus easy to follow by the readers. The validation experiments done by the authors are comprehensive and sufficient. There are only a few minor revisions/addendum:

L152, 159-163: There is some error in listing of quality control concentrations, the final concentrations do not add up from the working concentrations. It is suggested to keep the order of concentrations consistent, best listed in an ascending manner.

We thank the reviewer for finding the error in quality control concentrations reported in Line 152 that have been substituted with the correct ones (120, 25,5 µg/mL).

L301-304: The HPLC parameters changed from the method development (L186-189) to the plasma sample analysis. Please discuss why it has changed and the effects it brought. Normally, the method validated should be kept the same in GLP conditions.

The LC-MS/MS method applied for the quantification of IMA and norIMA in patients plasma sample was already in use in our laboratory (it is currently under revision by Journal of Pharmaceutical and Biomedical Analysis with the following title: “A New High-Performance Liquid Chromatography-Tandem Mass Spectrometry Method for Therapeutic Drug Monitoring of Imatinib and Norimatinib in GIST patients“). It was developed and validated according to FDA and EMA guidelines before the development of the assay for the quantification of DBS samples and thus it was used as REFERENCE method during the cross validation study (Line 294-295). DBS samples were analysed with the proposed method (COMPARANTOR method) while plasma samples were analysed with the reference method: the respective results were finally compared applying statistical analyses such as Passing-Bablok regression, Bland-Altman method, Lin’s concordance correlation coefficient and FDA/EMA requirements. 

L449: change “follow” to “follows”

Done.

---

## [Decision Letter · Decision Letter 1]

31 Oct 2019

Development and Validation of LC-MS/MS Method for Imatinib and Norimatinib Monitoring by Finger-prick DBS in Gastrointestinal Stromal Tumor Patients

PONE-D-19-17238R1

Dear Dr. Posocco,

We are pleased to inform you that your manuscript has been judged scientifically suitable for publication and will be formally accepted for publication once it complies with all outstanding technical requirements.

With kind regards,

John Matthew Koomen, PhD

Academic Editor

PLOS ONE

Additional Editor Comments (optional):

Reviewers' comments:

Reviewer's Responses to Questions

**Comments to the Author**

1. If the authors have adequately addressed your comments raised in a previous round of review and you feel that this manuscript is now acceptable for publication, you may indicate that here to bypass the “Comments to the Author” section, enter your conflict of interest statement in the “Confidential to Editor” section, and submit your "Accept" recommendation.

Reviewer #1: All comments have been addressed

Reviewer #2: All comments have been addressed

2. Is the manuscript technically sound, and do the data support the conclusions?

Reviewer #1: Yes

Reviewer #2: Yes

3. Has the statistical analysis been performed appropriately and rigorously? 

Reviewer #1: Yes

Reviewer #2: Yes

4. Have the authors made all data underlying the findings in their manuscript fully available?

Reviewer #1: Yes

Reviewer #2: Yes

5. Is the manuscript presented in an intelligible fashion and written in standard English?

Reviewer #1: Yes

Reviewer #2: Yes

6. Review Comments to the Author

Reviewer #1: Thank you for addressing my comments. You are correct that Line 373 already states the abbreviation and thus, does not need to be repeated. The change does make it easier for the reader.

Reviewer #2: The authors have addressed the reviewers' concerns satisfactorily and in my opinion, should proceed for acceptance.

7. PLOS authors have the option to publish the peer review history of their article (what does this mean?). If published, this will include your full peer review and any attached files.

Reviewer #1: Yes: Samer Sansil

Reviewer #2: No

---

## [Editor Report · Acceptance letter]

7 Nov 2019

PONE-D-19-17238R1 

Development and Validation of LC-MS/MS Method for Imatinib and Norimatinib Monitoring by Finger-prick DBS in Gastrointestinal Stromal Tumor Patients 

Dear Dr. Posocco:

I am pleased to inform you that your manuscript has been deemed suitable for publication in PLOS ONE. Congratulations! Your manuscript is now with our production department. 

With kind regards,

on behalf of

Dr. John Matthew Koomen 

Academic Editor

PLOS ONE